# The Discrepancy between Actual Performance and Self-Awareness among Adolescents with Executive Function Deficits

**DOI:** 10.3390/children9050684

**Published:** 2022-05-08

**Authors:** Yael Fogel

**Affiliations:** Department of Occupational Therapy, Ariel University, Ariel 4077625, Israel; yaelfo@ariel.ac.il; Tel.: +972-52-6262455

**Keywords:** adolescent health, executive function deficits, FITTED intervention, performance, real-life assessment

## Abstract

Adolescents with executive function deficits (EFD) struggle to perform complex daily activities and have difficulty being self-aware of their performance. This study aimed to compare actual performance with self-awareness of performance among adolescents with EFD before and after a metacognitive intervention. The participants consisted of 41 adolescents aged 10 to 14 years, previously diagnosed with EFD. All performed the Children’s Cooking Task (CCT), and completed the Behavioral Rating Inventory Executive Function—Self-Report (BRIEF-SR) and the Self-Awareness of Performance Questionnaire. Significant positive differences were found in the time duration and the total number of errors from the CCT and three BRIEF-SR subscale scores before and after the intervention. No significant differences were found in self-awareness of performance. After a cognitive intervention, adolescents with EFD improved their performance of a learned skill, but their self-awareness of their performance remained unchanged. These results may imply that EFD inhibits self-awareness development, and that self-awareness may not depend on task performance, but, rather, is influenced by other external factors. The article reports the secondary analysis from the results of the Functional Individualized Therapy for Teenagers with Executive Deficits (FITTED) intervention on human participants.

## 1. Introduction

Self-awareness is a complex, higher-level cognitive function that reflects a person’s ability to self-monitor, recognize and correct errors during a task [1]. It may also influence their ability to select appropriate task strategies [2]. Self-awareness develops gradually during childhood, beginning with the awareness of concrete aspects of behavior or physical characteristics, and graduating into more abstract concepts [3]. Reports have shown that children’s self-awareness increases with age, consistent with developing cognitive and linguistic skills [4]. Children show increased performance awareness from a young age and can better identify positive and negative aspects of their actual performance [5]. However, understanding the consequences of cognitive limitations in recognizing a problem when it occurs, or predicting a future problem, only develops in adolescence and young adulthood [6]. 

Adolescents with executive function deficits (EFD) differ from their peers with typical development (TD) in their struggle to complete daily life tasks [7,8,9], especially when environmental changes require them to adjust their thinking and actions [10]. Adolescents with EFD are characterized by disorganization, forgetfulness, the inability to multitask proficiently, and limitations in their ability to self-regulate behavior insightfully [11]. In turn, adolescents with EFD cope with limitations in daily functioning at home (e.g., day-to-day organizing, planning, and shifting focus), at school (e.g., learning ability and prioritizing responsibilities), and in social environments (e.g., understanding social situations and making friends) [12,13]. The literature has indicated that adolescents with EFD also have self-awareness deficits [14,15]. A lack of self-awareness regarding their differences with TD peers may negatively affect this group of adolescents [7,15]. Having no other coherent explanation for their functional difficulties, they may develop misattributions and negative self-beliefs [16].

Typically, self-awareness of impairment is evaluated by comparing a participant’s performance on neuropsychological tests and self-rating of cognitive performance [17,18]. Common impairment awareness tools include the Questionnaire of Executive Functioning [17], the Behavior Rating Inventory Executive Function (BRIEF), and its corresponding self-report, the BRIEF-SR [2,11]. 

Assessing self-awareness of performance is more complicated because it requires an evaluation of the discrepancy between actual and estimated performance during a specific activity [2]. Thus, its assessment cannot rely on self-reports alone. To the best of our knowledge, few studies have assessed self-awareness of performance among typical adolescents [2,19], let alone adolescents with EFD.

This study is a secondary analysis of data from a larger study on the effectiveness of a unique occupational therapy intervention, the Functional Individualized Therapy for Teenagers with Executive Deficits (FITTED), created to improve executive function in adolescents with EFD profiles. The FITTED is an 8-week, metacognitive, occupation-based program that aims to assist adolescents with EFD in improving performance and satisfaction with everyday life goals. An expanded explanation of that study can be found in [8]. 

This study compares the actual cooking task [20,21] performance with performance self-awareness before and after the task, prior to and following the FITTED intervention. We expected to find significant differences between pre- and postintervention scores in (a) actual performance of the cooking task, (b) self-awareness of the cooking task performance, and (c) self-awareness of EFD impairments. 

## 2. Materials and Methods

### 2.1. Participants

Study participants were recruited through community advertisements aimed at young adolescents with and without difficulty in daily functioning. We excluded participants with known psychiatric, emotional, or autism spectrum disorders; physical disabilities; or neurological diseases. This study presents a secondary analysis that includes 41 young adolescents (10–14 years) with EFD profiles who participated in the FITTED intervention [7]. Participants were characterized as having an EFD profile if their parent-reported scores were above the normal range (65 or higher) on the BRIEF behavioral regulation index (BRI) or metacognition index (MI). 

### 2.2. Procedure

The participating institution’s Ethics Committee approved this study (253/13), and all the adolescents and their parents signed informed consent prior to participation. In the primary study, those adolescents who met the inclusion criteria for the FITTED intervention were invited to individual sessions to complete the cooking task, which an expert occupational therapist administered and scored. Figure 1 presents the study design. The participants completed the BRIEF-SR and performed the Children’s Cooking Task (CCT) assessment pre- and postintervention. All participants completed the Self-Awareness of Performance Questionnaire (SAP-Q) before and after the CCT and FITTED intervention. 

### 2.3. Instruments

#### 2.3.1. BRIEF-SR

The BRIEF-SR [22] is a valid and reliable self-report instrument to assess executive function in 11- to 18-year-olds. Its 80 questions correlate to the BRIEF parent version in its four MI and four BRI subdomains. Adding the MI and BRI scores creates an overall global executive composite (GEC) score. Clinically significant t scores (M = 50 and SD = 10) are those that are 65 and above. The test–retest reliability of the BRI and MI were 0.84 and 0.87, respectively, and the internal reliability in the standardized sample was α = 0.80–0.98. The internal reliability of this study’s entire scale was α = 0.95. 

#### 2.3.2. CCT

The CCT is a performance-based evaluation [20,21] developed to assess executive function and multitasking abilities. It has high internal consistency (α = 0.81), moderate test–retest reliability for the total number of errors (0.65), and moderate concurrent validity with the BRIEF. It has been validated in Hebrew [7].

In the CCT, each participant is asked to follow two easy recipes: chocolate cake and fruit cocktail. Ingredients, utensils, and six recipes are laid on a table with an instruction sheet that shows the name of the dish, an ingredients list with illustrations, and numbered preparation steps with illustrations. Tasks are timed (min), and scores are classified into two error levels: descriptive and neuropsychological (to assess executive function and multitasking abilities). According to the CCT manual [23], these levels determine the number of errors by error type (descriptive), without reference to how or why they occurred; total errors (neuropsychological) allow a description of the reasons why each error occurred to be added. 

#### 2.3.3. SAP-Q 

This clinician-administered questionnaire is based on an instrument to assess the general awareness of performance [24,25], and is modified specifically for cooking performance tasks [26,27]. Before the task performance, the clinician asks participants three questions, which they rate from 1 (high estimation) to 5 (low estimation). These questions relate to performance (“How do you think you will do on the cooking task?”), expected difficulty (“Do you think you will have difficulty performing the cooking task?”), and estimated time (“How long do you think it will take you to perform the cooking task?”). After the task, participants are asked three more questions addressing the estimation of performance (“How do you think you did on the cooking task?”), satisfaction (“Are you satisfied with the way you performed the cooking task?”), and accuracy (“How accurately do you think you performed the cooking task?”). 

### 2.4. Data Analyses

The data were processed using SPSS 26. The sample did not distribute normally, so nonparametric tests were used. For the CCT and BRIEF-SR, Mann–Whitney tests were conducted to examine pre- and postintervention differences. Differences in the SAP-Q between the pre- and postintervention phases were analyzed using the Wilcoxon test for two related samples. Cohen’s d [28] was calculated for effect size, where 0.10 was considered a small effect, 0.30 a medium effect, and 0.50 a large effect.

Assessing self-awareness of performance requires an evaluation of the discrepancy between actual and estimated performance during a specific activity. Thus, new variables were calculated for the estimation before and after the CCT assessment: *Time estimation gap before and after* = estimated time after the CCT minus estimated time before the CCT;*Time estimation gap before and actual time duration in the CCT* = estimated time before the CCT minus actual time performing the CCT;*Time estimation gap after and actual time duration in the CCT* = estimated time after the CCT minus actual time performing the CCT.

## 3. Results

The participants included 29 boys (70.7%) and 12 girls (29.3%) with a mean age of 11.9 years (SD = 1.1). As an inclusion criterion, the parent BRIEF report scores equaled 65 and above (BRI: M = 67.70 and SD = 9.72; MI: M = 66.65 and SD = 6.34).

### 3.1. Pre- and Postintervention CCT Assessment Scores 

As shown in Table 1, significant differences were found between the pre- and postintervention scores in the actual CCT performance, including a decreased time duration (Z = −4.30; *p* < 0.001) and a reduction in the total number of performance errors (Z = −4.93; *p* < 0.001). 

### 3.2. Pre- and Postintervention BRIEF-SR Scores

Table 2 shows the significant differences found between the pre- and postintervention scores, regarding self-awareness of impairment, as measured by the BRIEF-SR GEC (Z = −2.29; *p* = 0.20) and MI (Z = −2.81; *p* = 0.005). No significant differences were found in the BRI (Z = −1.42; *p* = 0.15). However, significant differences were found in aspects of the BRI clinical scales, specifically, emotional control (Z = −2.31; *p* = 0.02) and monitor (Z = −2.06; *p* = 0.04). From the MI indices, significant differences were found in planning (Z = −2.40; *p* = 0.002), organization of materials (Z = −2.38; *p* = 0.02), and task completion (Z = −3.37; *p* = 0.001). 

### 3.3. Pre- and Postintervention SAP-Q Scores 

Although significant differences were found between the CCT assessing actual performance and the BRIEF-SR questionnaire (Table 3), only two SAP-Q items presented significant differences between pre- and postintervention: estimation of performance (Z = −2.127; *p* = 0.03) and time estimation (Z = −2.00; *p* = 0.04). Moreover, no significant differences were found in the variables time estimation gap before and after the cooking task (Z = −0.28; *p* = 0.77), time estimation gap before and actual time duration in the cooking task (Z = −1.33; *p* = 0.18), and time estimation gap after and actual time duration in the cooking task (Z = −1.52; *p* = 0.13). 

## 4. Discussion

This study emphasizes the discrepancy between actual performance and performance self-awareness among adolescents with EFD before and after a metacognitive intervention. As expected, significant differences were found in the cooking task assessment, indicating that the adolescents improved their performance greatly after completing the FITTED intervention. The participants reduced their time duration, total number of errors, and error types. Previous research on children with acquired brain injuries and severe dysexecutive syndrome [6] supports the improved task performance in our study. 

Previous studies have not reported differences in self-awareness of EFD through the BRIEF-SR questionnaire [29,30]. However, in this study, the BRIEF-SR scores showed significant differences pre- and postintervention in five scales: emotional control and monitor from the BRI, and plan, organization of materials, and task completion from the MI. These differences could mean that the adolescents’ self-awareness of their EFD did indeed change.

The FITTED intervention features supported the improvements in the CCT and BRIEF-SR in some scales. The FITTED intervention incorporates self-monitoring techniques with structured experience to assist adolescents in rediscovering themselves and redefining their knowledge of their strengths and weaknesses [8]. Such techniques may improve the adolescents’ ability to inhibit, self-regulate, and then respond to and channel self-directed executive actions. After the intervention, the participants paid more attention to the recipes, collected information more efficiently, and inhibited actions and reactions before performing the steps. They also adhered to the task sequence, added fewer unnecessary actions, succeeded in estimating amounts, and needed less assistance, as expressed by the decreased number of questions they asked [7]. 

As such, we expected—but did not find—a significant improvement in performance self-awareness and not just in actual performance. This lack of change prompts the following questions: Why did the intervention not affect the adolescents’ self-awareness of performance? Are they unaware of their ability to perform the task better? Do other components inhibit their ability to “see” and report their improvement? 

Three potential explanations are suggested to explain these unexpected results. First, adolescents with EFD are described as having impaired performance in complex daily living activities, requiring more ongoing assistance from adults, needing substantially more time to complete tasks, and engaging in far more dangerous activities than their TD peers [7,8,9]. Those difficulties may cause adolescents to become more distanced from the feedback they receive. Their difficulty in executing inhibition, using memory efficiently, exercising mental flexibility, and exhibiting self-control may delay the development of self-awareness. These characteristics could lead to them paying little attention to feedback from the environment and, thus, failing to integrate and update the self-knowledge necessary to develop self-awareness. Their neurological monitoring system, such as feedback, feedforward, and a comparative mechanism, may be damaged or impaired due to neurodevelopmental disorders or other health conditions that cause unawareness [31]. 

Second, adolescents are in a challenging period of development that includes comparing themselves to others while developing self-identity [32]. The combination of adolescence and living with EFD may affect their ability to cope, progress, and become more self-aware [33]. This may lead to various forms of unawareness, resulting from psychologically motivated denial [31]. This denial is a coping mechanism that people create as protection from a painful reality or from recognizing distressing aspects of themselves in the face of adversity [34]. Denial can prevent people from acquiring effective coping skills and developing realistic goals [35,36]. Adolescents’ choices to deny their skills and challenges seem understandable and may serve as a protective strategy from personal failure [37,38].

A third explanation could be that adolescents with EFD profiles experience years of struggle, particularly in filling the gap between the external and their own environments [12]. Adults tend to misunderstand EFD performance issues among adolescents and refer to their externalizing behavior as lazy, lacking motivation, or willful misbehavior [39]. Those adolescents may receive harmful feedback, which may influence their self-awareness [31]. According to Toglia and Kirk [40], subjective cognitive abilities are based mainly on subjective feelings of effort and failure. These beliefs may impede their ability to develop healthy and adaptive self-awareness.

## 5. Conclusions

Actual performance and self-awareness of executive function impairment improved following the metacognitive intervention, but self-awareness of performance did not. Self-awareness of performance is not an automatically developed process, it is a skill that requires nurturing and development [6]. Clinically, there is a need to consider self-awareness in the evaluation process. If viewed as a significant therapy goal, self-awareness can strengthen the ability of adolescents with EFD to self-monitor, recognize, and correct errors during a task, and select appropriate task strategies. Additionally, improving awareness of specific task performance may take longer than improving the actual performance. Thus, there is a need to train, practice, and build many experiences for children with EFD to help them develop increased self-awareness.

Theoretically, this study provides additional evidence highlighting this population’s complexity. We found improved performance and achievement in daily function goals, but the adolescents’ self-awareness of their performance stayed the same. These adolescents need continued follow-up, even after completing the treatment process. It may be assumed that their awareness is not always task-dependent, and more components are involved.

This study leaves unanswered questions and underscores the need for further research. We tested self-awareness using questions before and after performing a cooking performance task. It is crucial to examine the findings of other performance tasks related to adolescent daily functioning, such as writing, play, and social participation activities, to understand whether self-awareness of performance is task-dependent and consistent, even when performance has improved. Further, we analyzed self-awareness of performance in only one way. It is necessary to assess self-awareness of performance using different tools to verify the reliability of the self-awareness questionnaire. Moreover, other well-known factors that contribute to EFD, such as depression and anxiety, may not have been taken into account in the current study. 

Follow-up studies should examine factors such as adolescent self-awareness over time, changes with age in adolescence as a variable, and results with and without therapeutic intervention, as well as referring to mental and emotional components that relate to adolescence, with tools such as the Behavior Assessment System for Children (BASC) [41]. Additional components related to the adolescent’s environment, such as parental attitudes, educational frameworks, the adolescent’s developmental and medical history, and emotional elements that may affect self-awareness, should be examined.

## Figures and Tables

**Figure 1 children-09-00684-f001:**
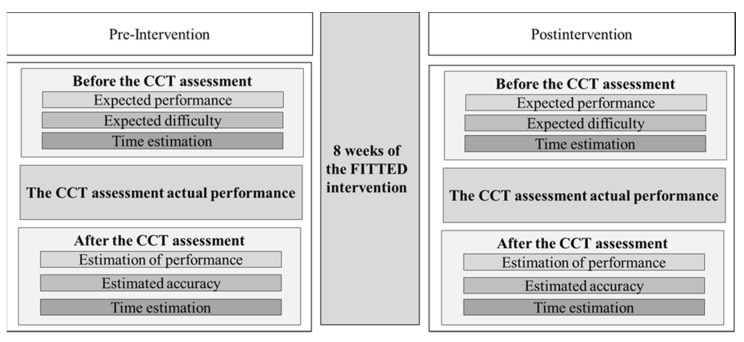
Study design.

**Table 1 children-09-00684-t001:** Comparison of pre- and postintervention Children’s Cooking Task scores (performance).

Variable	Pre-Intervention	Postintervention	Z	*p*	*d*
	M (SD) [min–max]				
**Task duration (min)**	27.87 (12.22) [11–75]	20.24 (5.90) [11–37]	−4.30	<0.001	0.79
**Total number of errors**	75.98 (52.98) [15–290]	41.22 (24.83) [6–202]	−4.93	<0.001	0.84
**Descriptive analysis**					
**Omissions**	2.54 (1.88) [0–8]	1.46 (1.25) [0–5]	−3.13	0.002	0.68
**Additions**	13.22 (9.15) [1–51]	4.95 (3.97) [0–15]	−4.80	<0.001	1.17
**Substitutions—sequences**	5.32 (3.08) [0–11]	3.12 (2.49) [0–9]	−3.04	0.002	0.78
**Estimation errors**	5.63 (2.40) [0–10]	2.98 (2.63) [0–10]	−4.27	<0.001	1.05
**Commentary—question**	18.88 (20.54) [0–113]	4.83 (7.69) [0–33]	−4.34	<0.001	0.90
**Neuropsychological analysis**					
**Control errors**	7.80 (4.56) [1–21]	4.59 (3.26) [0–14]	−3.69	<0.001	0.81
**Context neglect**	31.27 (21.24) [8–120]	12.27 (8.82) [1–37]	−4.89	<0.001	1.17
**Environmental adherence**	5.20 (3.87) [0–23]	2.37 (1.70) [0–7]	−4.31	<0.001	0.95
**Purposeless action**	6.22 (5.65) [0–20]	2.46 (2.30) [0–8]	−3.52	<0.001	0.87
**Dependency**	5.58 (4.18) [0–17]	1.39 (1.98) [0–8]	−4.72	<0.001	1.28
**Inappropriate behavior**	0.63 (1.09) [0–4]	0.34 (0.82) [0–8]	−1.16	0.246	0.30

**Table 2 children-09-00684-t002:** Comparison of pre- and postintervention self-awareness of executive functions (awareness of impairment).

Scale	Pre-Intervention	Postintervention	Z	*p*	*d*
	M (SD) [min–max]				
**BRIEF-SR BRI**	51.08 (12.18) [37–84]	56.82 (11.17) [35–84]	−1.42	0.150	0.49
**BRIEF-SR MI**	58.69 (10.83) [31–81]	55.46 (9.52) [32–78]	−2.81	0.005	0.32
**BRIEF-SR GEC**	59.41 (11.24) [33–84]	56.59 (10.86) [33–84]	−2.29	0.020	0.25
**BRIEF-SR scales**					
**Inhibition**	55.69 (11.08) [34–86]	54.28 (10.47) [34–79]	−0.78	0.440	0.32
**Shift**	58.59 (14.36) [32–91]	57.33 (13.12) [34–97]	−0.74	0.46	0.10
**Emotional control**	61.18 (10.96) [38–83]	57.74 (10.37) [38–82]	−2.31	0.020	0.32
**Monitor**	54.10 (10.10) [36–76]	51.02 (10.70) [36–78]	−2.06	0.040	0.30
**Working memory**	56.05 (11.39) [34–86]	55.41 (10.75) [34–81]	−5.01	0.610	0.06
**Plan/Org**	58.33 (10.89) [31–79]	55.49 (9.81) [31–77]	−2.40	0.020	0.27
**Organization of materials**	55.23 (11.55) [33–76]	51.64 (8.41) [33–75]	−2.38	0.020	0.35
**Task completion**	61.13 (11.00) [35–84]	56.33 (8.60) [35–72]	−3.37	0.001	0.49

*Note*. BRIEF-SR = Behavioral Rating Inventory Executive Function—Self-Report; BRI = behavioral regulation index; MI = metacognition index; GEC = global executive composite.

**Table 3 children-09-00684-t003:** Comparison of pre- and postintervention Children’s Cooking Task (CCT) scores.

Variable	Pre-Intervention	Postintervention	*Z*	*p*
	M (SD) [min–max]			
Before the CCT				
Expected performance	3.32 (1.17) [1–5]	3.46 (1.03) [1–5]	−0.92	0.33
Expected difficulty	3.54 (0.98) [2–5]	3.78 (0.941) [1–5]	−1.55	0.12
Time estimation	32.32 (20.40) [5–90]	27.12 (11.71) [10–60]	−0.82	0.41
After the CCT				
Estimation of performance	3.29 (1.03) [1–5]	3.76 (0.94) [1–5]	−2.13	0.03
Estimated accuracy	3.78 (0.94) [1–5]	4.10 (0.77) [2–5]	−1.71	0.09
Time estimation	26.78 (12.46) [10–60]	21.83 (10.58) [5–60]	−2.00	0.04

## Data Availability

The datasets generated and/or analyzed during the current study are not publicly available due to ethical restrictions, but are available from the corresponding author on reasonable request.

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
