# Peer review of "The Discrepancy between Actual Performance and Self-Awareness among Adolescents with Executive Function Deficits"

_children, 2022, doi:10.3390/children9050684_

Round 1

Reviewer 1 Report

Requires minor revisions: assumes the text is almost ready for publication except for minor details:
o There are minor spelling or syntactical errors.
o Some statements are not referenced
o Bibliographical references are too short.
o Some secondary aspect of the text that does not affect the key conclusions of the article needs to be elucidated key conclusions of the article.

Author Response

Replies to the comments of the reviewers

Thank you for the opportunity to revise the manuscript. Based on the Editor's and reviewers' comments, I have substantially restructured and revised the manuscript.

Reviewer: 1

Comments for the author

There are minor spelling or syntactical errors.

A language editor went over the article again and made some changes, for example, correcting split infinitives and spelling out “years” rather than using the SI unit abbreviation.

Some statements are not referenced

Thank you for this comment. The author went through the article and added seven new references (1, 14, 15, 18, 33–35) and multiple new citations for the statements.

Bibliographical references are too short.

Assuming this comment means there were too few references, the reference list gets longer. As noted in the previous comment, there are currently 42 reference sources. The references are listed and formatted according to the journal’s template and guidelines (e.g., abbreviated journal titles).

Some secondary aspect of the text that does not affect the key conclusions of the article needs to be elucidated key conclusions of the article.

Thank you for this comment. To the best of my understanding, this comment is saying that some results were not reflected in the study's conclusions. I have reorganized the Conclusion section. It now opens with a statement addressing the secondary aspects: “Actual performance and self-awareness of executive function impairment improved following the metacognitive intervention, but self-awareness of performance did not. Self-awareness of performance is not an automatically developed process; it is a skill that needs nurturing and development [6].”

Reviewer 2 Report

Thank you for allowing me to review this article. It is an important topic but will require depth to a complex problem. The conclusion starts off with saying it is a complex issue but is not clear on whether the study helps us move closer to it.

Given the depressive and anxiety symptoms contributing to EF issues, please explain why the BASC was not used: Reference:” the BRIEF does not appear to tap into internalizing disorders to the same extent as the BASC. “
Assessment of Attention Deficit Hyperactivity Disorder (ADHD) Using the BASC and BRIEF Kelly Pizzitola Jarratt, Cynthia A. Riccio, and Becky M. Siekierski Department of Educational Psychology, Texas A&M University,

Will the authors talk about how self awareness changes with age in adolescents as a variable ? https://www.ncbi.nlm.nih.gov/pmc/articles/PMC548185/

Author Response

Replies to the comments of the reviewers

Thank you for the opportunity to revise the manuscript. Based on the Editor's and reviewers' comments, I have substantially restructured and revised the manuscript.

Reviewer: 2

Comments for the author

The conclusion starts off with saying it is a complex issue but is not clear on whether the study helps us move closer to it.

Thank you for this comment. Yes, it helps us move closer but there is still a long way to go. I have rewritten/reorganized the Conclusion section and deleted that first sentence, instead addressing some secondary aspects (see response to Reviewer 1).

Given the depressive and anxiety symptoms contributing to EF issues, please explain why the BASC was not used: Reference:” the BRIEF does not appear to tap into internalizing disorders to the same extent as the BASC. “
Assessment of Attention Deficit Hyperactivity Disorder (ADHD) Using the BASC and BRIEF Kelly Pizzitola Jarratt, Cynthia A. Riccio, and Becky M. Siekierski Department of Educational Psychology, Texas A&M University,

Thank you for this comment. I was unaware of that questionnaire. The BRIEF questionnaire responds to the EF difficulties profile and not situations such as depression or anxiety.

I added in the study's limitations that information about these situations using questionnaires like the BASC might contribute to understanding the ability to report managerial functions. Moreover, in further research on the subject, one should be careful about that.

I added statements to reflect this and your points regarding mental/emotional components that contribute to EFD:

“Moreover, other well-known factors that contribute to EFD, such as depression and anxiety-ty, may not have been taken into account in the current study.

“Follow-up studies should examine adolescent self-awareness over time, changes with age in adolescence as a variable, and results with and without therapeutic intervention and refer to mental and emotional components that relate to adolescence with tools such as the Behavior Assessment System for Children (BASC) [42].”

Will the authors talk about how self awareness changes with age in adolescents as a variable ? https://www.ncbi.nlm.nih.gov/pmc/articles/PMC548185/

Thank you for the critical comment. In this analysis, I did not focus on changes during the adolescent period but can contribute in the future to examining this specifically. I have added this as a recommendation in the future study section.